# Quantitative stratigraphic analysis in a source-to-sink numerical framework

Xuesong Ding[1], Tristan Salles[1], Nicolas Flament[2], and Patrice Rey[1]

[1]Basin GENESIS Hub, EarthByte Group, School of Geosciences, The University of Sydney, Sydney, NSW 2006, Australia
[2]School of Earth and Environmental Science, University of Wollongong, Wollongong, NSW 2522, Australia

**Correspondence:** Xuesong Ding (xuesong.ding@sydney.edu.au)

**Abstract.**

The sedimentary architecture at continental margins reflects the interplay between the rate of change of accommodation creation ($\delta A$) and the rate of change of sediment supply ($\delta S$). Stratigraphic interpretation increasingly focuses on understanding the link between deposition patterns and changes in $\delta A/\delta S$, with an attempt to reconstruct the contributing factors. Here, we use the landscape modelling code *pyBadlands* to (1) investigate the development of stratigraphic sequences in a source-to-sink context; (2) assess the respective performance of two well-established stratigraphic interpretation techniques: the trajectory analysis method and the accommodation succession method; and (3) propose quantitative stratigraphic interpretations based on those two techniques. In contrast to most Stratigraphic Forward Models (SFMs), *pyBadlands* provides self-consistent sediment supply to basin margins as it simulates erosion, sediment transport and deposition in a source-to-sink context. We present a generic case of landscape evolution that takes into account periodic sea level variations and passive margin thermal subsidence over 30 million years, under uniform rainfall. A set of post-processing tools are provided to analyze the predicted stratigraphic architecture. We first reconstruct the temporal evolution of the depositional cycles and identify key stratigraphic surfaces based on observations of stratal geometries and facies relationships, which we use for comparison to stratigraphic interpretations. We then apply both the trajectory analysis and the accommodation succession methods to manually map key stratigraphic surfaces and define sequence units on the final model output. Finally, we calculate shoreline and shelf-edge trajectories, the temporal evolution of changes in relative sea level (proxy for $\delta A$) and sedimentation rate (proxy for $\delta S$) at the shoreline, and automatically produce stratigraphic interpretations. Our results suggest that the analysis of the presented model is more robust with the accommodation succession method than with the trajectory analysis method. Stratigraphic analysis based on manually extracted shoreline/shelf-edge trajectory requires calibrations of time-dependent processes such as thermal subsidence or additional constraints from stratal terminations to obtain reliable interpretations. The 3D stratigraphic analysis of the presented model reveals small lateral variations of sequence formations. Our work provides an efficient and flexible quantitative sequence stratigraphic framework to evaluate the main drivers (climate, sea level and tectonics) controlling sedimentary architectures and investigate their respective roles in sedimentary basins development.

# 1 Introduction

Since its introduction in 1970's, sequence stratigraphy has been widely used to interpret depositional architectures in terms of variations in eustatic sea level or relative sea level (*i.e.* accommodation) (Vail et al., 1977a; Pitman, 1978; Posamentier et al., 1988; Posamentier and Vail, 1988; Jervey, 1988). With recognition of the role of sediment supply in affecting stratal stacking patterns, the rate of change of accommodation creation ($\delta A$) versus the rate of change of sediment supply ($\delta S$) - the $\delta A/\delta S$ ratio - has been widely accepted as the main control of sequence formations (Schlager, 1993; Muto and Steel, 1997; Catuneanu et al., 2009; Neal and Abreu, 2009; Neal et al., 2016). The $\delta A/\delta S$ concept offers several advantages compared to conventional stratigraphic models as it directly relates depositional patterns to the main contributing geological drivers (*e.g.* eustasy, tectonics and sediment supply). Yet, the inherent difficulties in accurately describing accommodation and reconstructing sediment supply limits the quantification of the $\delta A/\delta S$ ratio and, as a result, the practical application of the $\delta A/\delta S$ concept in stratigraphic interpretations (Muto and Steel, 2000; Burgess et al., 2016).

Experimental stratigraphy (e.g. analogue experiments, stratigraphic forward modelling (SFM)) plays a significant role in exploring the development of sedimentary architecture under various forcing conditions (Martin et al., 2009; Burgess et al., 2012; Muto et al., 2016). Over the past few decades, SFM has been widely used to investigate the interplay between major sequence-controlling factors (e.g. eustasy, tectonics, flexural isostasy, sediment supply, sediment compaction, basin physiography, etc.) and their influences in the formation of stratigraphic sequences (Reynolds et al., 1991; Posamentier and Allen, 1993; Steckler et al., 1993; Carvajal and Steel, 2009; Burgess et al., 2012; Granjeon et al., 2014; Csato et al., 2014; Sylvester et al., 2015; Harris et al., 2015, 2016). SFM can therefore provide insights into the quantification of links between sequence formation and the changing $\delta A/\delta S$ ratio. Here we use SFM as a tool to investigate the development of stratigraphic architecture under the interplay between accommodation variations and sediment supply. We use the landscape evolution code *pyBadlands* that describes sediment transport from source to sink in a self-consistent manner (Salles, 2016; Salles and Hardiman, 2016; Salles et al., 2018). In *pyBadlands*, the erosion from upstream catchments is linked to the sedimentation on basin margins through sediment routing resulting from a combination of channelling and hillslope processes. As a consequence, sediment supply to basin margins is dynamically determined without user control; it results from the interaction of imposed tectonics, climatic and eustatic variations as well as autogenic changes in upstream catchment physiography.

We then apply the rules of sequence stratigraphy to interpret predicted depositional cycles. In sequence stratigraphy, various sequence models exist and subdivide the stratal successions into unconformity and/or correlated conformity bounded stratigraphic units (Galloway, 1989; Mitchum Jr, 1977; Van Wagoner et al., 1988; Helland-Hansen and Gjelberg, 1994). These models have been proved to be useful for a number of cases. However, the interpretation of systems tracts and sequences based on these sequence models can be non-objective and non-unique (Burgess et al., 2016; Burgess and Prince, 2015). Observation-based methods to interpret stratigraphic sequences have the advantage of being objective and independent of spatial and temporal scales. Hence, over the years, it has been recognised that stratigraphic interpretations should be guided by physical observations, and be independent of depositional models and associated assumptions (Abreu et al., 2014; Burgess et al., 2016; Neal et al., 2016). Here, we focus on two such methods: the trajectory analysis and the accommodation succession method.

Helland-Hansen et al. (1994; 1996; 2009) proposed the trajectory analysis method that correlates depositional units with the lateral and vertical migrations of the shoreline and shelf-edge trajectories. Neal et al. (2009; 2016) proposed a refined sequence stratigraphic framework known as the accommodation succession method in which sequence sets are interpreted based on offlap break trajectory and stratal geometries. The temporal evolution of accommodation change and sediment supply can then be derived from interpreted sequence sets and key stratigraphic bounding surfaces.

This study also attempts to evaluate the performance of these two approaches to interpret the stratal architecture predicted with *pyBadlands*. To illustrate the workflow, we build a synthetic source-to-sink model that includes a mountain range (sediment source), an alluvial plain (sediment transfer zone) and a passive continental margin (sink area) where relatively well understood sequence-controlling drivers such as eustasy and thermal subsidence (Watts and Steckler, 1979; Bond et al., 1989; Steckler et al., 1993) are imposed. We first present the temporal evolution of predicted stratal stacking patterns and map out key stratigraphic surfaces, which serves as a reference for comparison to stratigraphic interpretations. We then follow the trajectory analysis and the accommodation succession method to interpret predicted stratal architecture. Finally, we design a suite of numerical tools to extract of shoreline and shelf-edge trajectories, as well as accommodation change and sedimentation evolution over space and time, with the aim to integrate the trajectory analysis method and the accommodation succession method within *pyBadlands* to derive quantitative interpretations. These new capabilities make it possible to quickly interpret synthetic depositional cycles in a consistent manner using either the trajectory analysis or the accommodation succession method.

## 2   Quantitative stratigraphic analysis in *pyBadlands*

The workflow to build a quantitative framework of stratigraphic analysis in *pyBadlands* is illustrated in Figure 1. In this study, we focus on the post-processing of model outputs. *pyBadlands* records the depth, relative elevation (depth relative to sea level) and thickness of each stratigraphic layer through time. With this information, we are able to extract cross-sections and to reconstruct the temporal evolution of stratal stacking patterns and 3D Wheeler diagrams at any locations. The 3D Wheeler diagram contains information of distance along the cross-section, time and deposited sediment thickness. This allows us to identify key stratigraphic surfaces based on observations of stratal geometry and facies relationships.

We then interpret the synthetic depositional cycles in two ways. First, we follow the workflow proposed in the trajectory analysis and the accommodation succession method to subdivide the stratigraphic record. The trajectory analysis technique defines different trajectory classes based on the trajectories recorded at either shoreline or shelf-edge positions (Helland-Hansen and Gjelberg, 1994; Helland-Hansen and Martinsen, 1996; Helland-Hansen and Hampson, 2009). Though both represents a break-in-slope, the shelf-edge evolves at larger spatial (Fig. 2) and over longer temporal scales than the shoreline, making it easier to define the shelf-edge on seismic data. By investigating the migration direction of the shoreline and the shelf-edge, four shoreline trajectory classes and three shelf-edge trajectory classes are used to characterise the successive depositional packages. The shoreline trajectory classes include the transgressive trajectory class (TTC), the ascending regressive trajectory class (ARTC), the descending regressive trajectory class (DRTC) and the stationary (*i.e.* non-migratory) shoreline trajectory class.

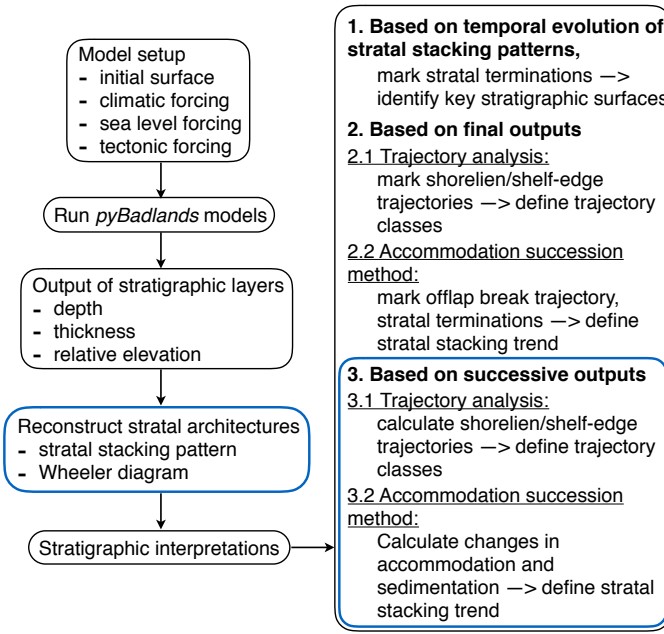

**Figure 1.** Workflow to build the stratigraphic architecture in *pyBadlands*. We present three different ways of stratigraphic interpretation. The new post-processing workflows designed to automatically interpret stratigraphic sequences are shown in the blue boxes.

The shelf-edge trajectory classes include descending trajectory class (DTC), ascending trajectory class (ATC), transgressive trajectory class (T) and stationary or flat trajectory class (Fig. 2).

Neal et al. (2009; 2016) presented a refined hierarchy framework, known as the accommodation succession method, in which the subdivision of depositional units are entirely based on the stratal geometry resulting from the evolution of accom-

modation change and sediment infill. Three stacking patterns and their bounding surfaces are defined and subsequently used to assess the changing history of accommodation and sediment supply (Fig. 2). These include the retrogradation stacking (R) for $\delta A/\delta S > 1$, the progradation to aggradation stacking (PA) for $\delta A/\delta S < 1$ and increasing, and the aggradation to progradation (even to degradation) stacking (AP or APD) for $\delta A/\delta S < 1$ and decreasing (and possibly negative). The three key surfaces bounding these stacking patterns are the sequence boundary (SB), the maximum transgressive surface (MTS) and the maximum

regressive surface (MRS).

We manually mark the shelf-edge (or offlap break) trajectory and stratal terminations on the final output of stratal stacking pattern reconstructed from a simulated cross-section. Key stratigraphic surfaces and stacking patterns are then defined.

Second, a series of post-processing tools are implemented in *pyBadlands* to numerically extract the shoreline and shelf-edge positions, and the temporal evolution of $\delta A$ and $\delta S$ through time and space (Fig. 3). The shoreline position is recorded by

tracking the topographic contour that corresponds to sea level. The shelf-edge is calculated by assuming a critical slope of 0.025 degrees in this case. Changes in relative sea level and sedimentation rate are used as proxies for $\delta A$ and $\delta S$ (Poag and Sevon, 1989; Galloway and Williams, 1991; Liu and Galloway, 1997; Galloway, 2001). Therefore, $\delta A$ at any given location

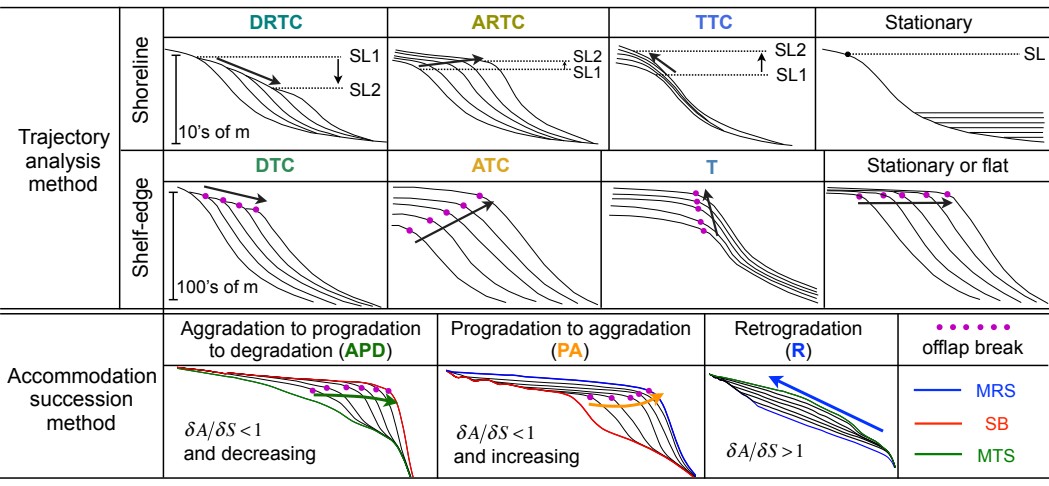

**Figure 2.** Stratigraphic sequence interpretation approaches used in this study. In the trajectory analysis method, four shoreline trajectory classes and three shelf-edge trajectory classes are delineated based on the lateral and vertical migrations of shoreline and shelf-edge. The shoreline trajectory classes include descending regressive trajectory class (DRTC), ascending regressive trajectory class (ARTC), transgressive trajectory class (TTC) and stationary trajectory class. The shelf-edge trajectory classes include descending (DTC), ascending (ATC), transgressive (T) and stationary shelf-edge trajectory classes (Helland-Hansen and Gjelberg, 1994; Helland-Hansen and Martinsen, 1996; Helland-Hansen and Hampson, 2009). In the accommodation succession method, three types of stacking patterns and their bounding surfaces are defined based on observations of stratal terminations (e.g. onlap, offlap, etc) and stratal geometries, including aggradation to progradation to degradation (APD) stacking, progradation to aggradation (PA) stacking and retrogradation (R) stacking. The bounding surfaces are sequence boundary (SB), maximum transgressive surface (MTS) and maximum regressive surface (MRS). Each stacking pattern reflects changes in the rate of accommodation creation ($\delta A$) with respect to the rate of sediment supply ($\delta S$) at the shoreline. APD stacking corresponds to $\delta A/\delta S < 1$, with a decreasing trend that can be negative; PA stacking represents $\delta A/\delta S < 1$ and increasing; finally, R stacking occurs for $\delta A/\delta S > 1$ (Neal and Abreu, 2009; Neal et al., 2016).

from time T1 to T2 integrates changes in eustatic sea level and basement subsidence, $\delta S$ at any given location between time T1 and T2 is given by deposited strata thickness. In this study, both $\delta A$ and $\delta S$ are in m/Myr. We then extract $\delta A$ and $\delta S$ at shoreline positions through time. Trajectory classes, stacking patterns and stratigraphic surfaces are defined automatically based on calculated shoreline, shelf-edge trajectories and time-dependent $\delta A$ and $\delta S$.

## 3 Experimental setup

We provide an example to illustrate how our workflow can be used to automatically generate stratigraphic sequences and analyse them in an integrated numerical toolbox.

We create an initial synthetic landscape of dimensions 300 km by 200 km with a spatial resolution of 0.5 km. The region includes a mountain range, an alluvial plain and an adjacent continental margin consisting of a continental shelf, a continental

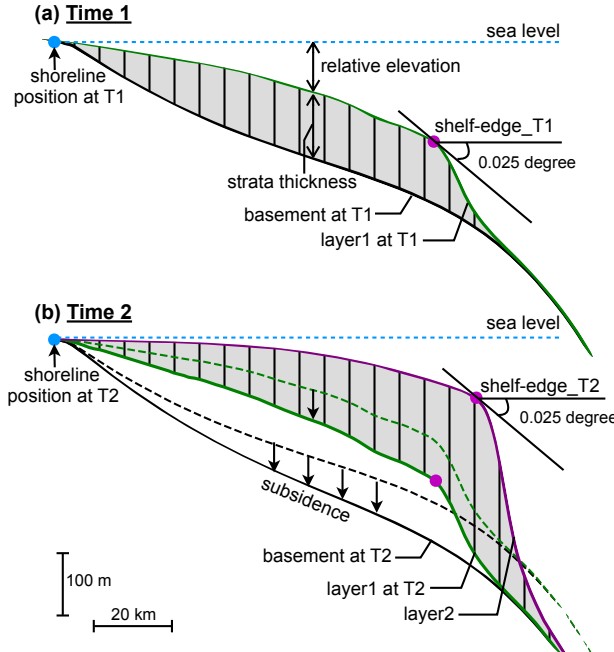

**Figure 3.** Schematic diagram showing the sedimentation from Time 1 (T1) to Time 2 (T2) under sea level variation and basement subsidence. The depth, relative elevation and thickness of stratigraphic layer are recorded at every time step (dT = T2-T1). Post-processing tools extract the shoreline and shelf-edge positions, and calculate the rate of accommodation creation ($\delta A$) and the rate of sedimentation ($\delta S$) through time. The shoreline position is recorded by tracing the topographic contour that corresponds to sea level. The shelf-edge is calculated by assuming a critical slope 0.025 degree. $\delta A$ is calculated as changes in relative sea level at shoreline position over (T2-T1): (sea level change + subsidence)/(T2-T1); $\delta S$ is calculated as deposited strata thickness at shoreline position over (T2-T1): (strata thickness)/(T2-T1).

slope and an oceanic basin. Details of the geometry are presented in Figure 4a. The model duration is 30 Myr, generating 300 stratigraphic layers with display intervals every 0.1 Myr. Our model setting mimics the first-order, long-term landscape evolution and associated stratigraphic sequence development along a passive continental margin, with forcing conditions including climate, eustatic sea level change and thermal subsidence.

5    In this study, we use a single flow direction, detachment-limited stream power law, and a simple downslope creep law to describe erosion, sediment transport and deposition. Equations and model parameters are provided in the Supplementary material. Considering that this study focuses on long-term stratigraphic evolution due to sea level change, both climate and subsidence patterns are considered constant. Climate is assumed to be directly related to precipitation with a spatially and temporally uniform precipitation rate of 2.0 m/yr over 30 Myr. The sediment input varies through time, depending on the

10   dynamic evolution of source area.

    Sea level fluctuations are a major driver of changes in accommodation and thus stratigraphic sequence development. They act at different temporal scales, resulting in various stratigraphic cycles with first-order cycles of duration around 200-300

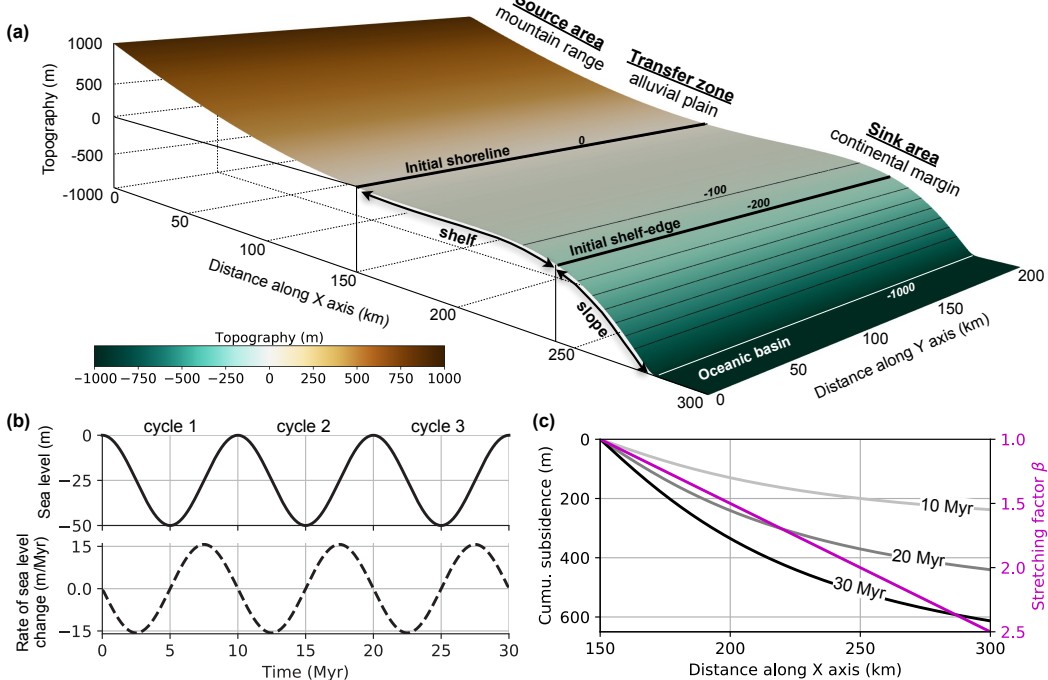

**Figure 4.** Model setup. (a) The initial surface elevation of the mountain region ranges from 200 m to 1000 m over a width of 100 km, while the elevation of the alluvial plain ranges from 0 m to 200 m over a width of 50 km. The sink area includes a continental margin in which the elevation of the continental shelf ranges from -250 m to 0 m over a width of 80 km, the elevation of the continental slope ranges from -1000 m to -250 m over a width of 40 km and a flat oceanic basin whose depth is -1000 m over a width of 30 km. Black lines are isobaths with an interval of 100 m. The initial elevations of shoreline and shelf-edge are 0 m and -200 m. (b) Eustatic sea level and its rate of change over 30 Myr. The eustatic sea level scenario modelled using a sinusoidal curve consisting of three 10 Myr cycles of peak-to-peak amplitude 50 m. (c) Distance-dependent stretching factor $\beta$ and the resulting thermal subsidence at 10 Myr, 20 Myr, and 30 Myr across the continental margin.

Myr; second-order cycles of duration around 10-80 Myr; and third-order cycles of duration 1-10 Myr (Vail et al., 1977b). Here we consider the effects of long-term eustatic cycles and assume that eustasy is independent of climate and local tectonics. For simplicity, eustatic sea level is modelled using a sinusoidal curve consisting of three 10 Myr cycles of 50 m amplitude (Fig. 4b), which correspond to second- to third-order eustatic fluctuations (Vail et al., 1977b).

5     Thermal subsidence is an important process leading to the deepening of basin floor due to isostatic adjustment during lithosphere cooling. A simple stretching model from McKenzie (1978) is applied in this study, in which subsidence is produced by thermal relaxation following an episode of extension. The equation to derive the subsidence at time $t$ is:

$$S(t) = E_0 r - E_0 exp(-t/\tau) \tag{1}$$

where $E_0 = 4a\rho_0\alpha T_1/\pi^2(\rho_0 - \rho_w)$, $r = \frac{\beta}{\pi}sin(\frac{\pi}{\beta})$, $a = 125$ km is the thickness of lithosphere, $\rho_0 = 3300$ kg/m$^3$ the density 10  of the mantle at 0°C, $\rho_w = 1000$ kg/m$^3$ the density of seawater, $\alpha = 3.28 \times 10^{-5}$ K$^{-1}$ the thermal expansion coefficient

for both the mantle and the crust, $T_1 = 1333^o$ C the temperature of the asthenosphere, and $\tau = 62.8$ Myr the characteristic thermal diffusion time. The stretching factor $\beta$ characterises the extension of the lithosphere. These parameters are taken from McKenzie (1978). In our experiments, thermal subsidence is imposed on the continental margin, starting from the initial shoreline (Fig. 4a), which experiences the least subsidence, to the outward edge where subsidence is maximum. We take $\beta$ as distance-dependent and calculate the thermal subsidence accumulated at 10 Myr, 20 Myr and 30 Myr, respectively (Fig. 4c). The subsidence rate is constant at each single position but increases along the X-axis. In our model, relative sea level is the combined result of eustatic sea level and thermal subsidence, and thus varies spatially due to basement subsidence.

## 4 Results

### 4.1 Temporal evolution of stratal architecture

Our post-processing tools quickly extract simulated stratal stacking patterns as well as Wheeler diagrams in any region of the simulated domain. Figure 5 presents the development of stratal stacking pattern generated along a dip-oriented cross-section through the middle of the domain. The stratal architecture is coloured based on depositional environments defined using six paleo-depth windows. We infer depositional facies, changes in depositional trends, stratal terminations and stratal geometries from the temporal evolution of stratal stacking patterns. These information is then used to identify stratigraphic surfaces and to define systems tracts and sequences (Van Wagoner et al., 1988). Sediment flux is extracted along the cross-section by computing the total volume of deposited sediments per unit width in 0.5 Myr intervals (Fig. 5d). As showed in Figure 5a, an oblique prograding clinoform develops due to the falling eustatic sea level, with strata toplapped by a subaerial unconformity (SU). This subaerial unconformity terminates downdip at the offlap-break at 4.0 Myr and it transfers to a marine correlative conformity (CC$^*$), which together forms a sequence boundary (Fig. 5a). The shoreline steps back and strata onlap the subaerial unconformity from 4.0 Myr. The prograding packages then shift to aggradational pattern until 6.5 Myr when the shelf-edge reaches its most basinward location, marked by the maximum regressive surface (MRS) at 6.5 Myr. Eustatic sea level first falls and then rises at a relative slow changing rate between 4.0 Myr and 6.5 Myr and the clinoform formed during this period is characterised by thin topsets and thick foresets. The sediment flux constantly increases from 0.7 km$^2$/Myr to 2.0 km$^2$/Myr over the first 6.5 Myr (Fig. 5d), with small variations due to the lateral shifts in the position of river mouth. Eustatic sea level rises from 6.5 Myr to 9.0 Myr, and strata continues onlapping the subaerial unconformity and fills incised channels with fluvial sediments to form a maximum flooding surface (MFS) at 9.0 Myr. The clinoform formed during this time period is characterised by thick topsets and an absence of foresets. From 9.0 Myr to 10.0 Myr, the shoreline steps back and strata continues onlapping while the stacking pattern changes from retrogradation to aggradation. During the phase of sea level rise from 6.5 Myr to 10.0 Myr, the sediment flux slightly decreases to 1.6 km$^2$/Myr (Fig. 5d). Eustatic sea level falls from 10 Myr to 13.5 Myr and strata stacking changes to progradation, forming a second subaerial unconformity (Fig. 5b). The surface formed at 10.0 Myr that separates aggradation from progradation is defined as correlative conformity (CC). Mid-slope sediments start to accumulate within the progradational clinoform between 10.0 Myr and 13.5 Myr. The sediment flux from 10.0 Myr to 13.5 Myr reveals a gentle increasing trend with large variations of up to 0.8 km$^2$/Myr, followed by a significant drop in sediment flux at 13.5

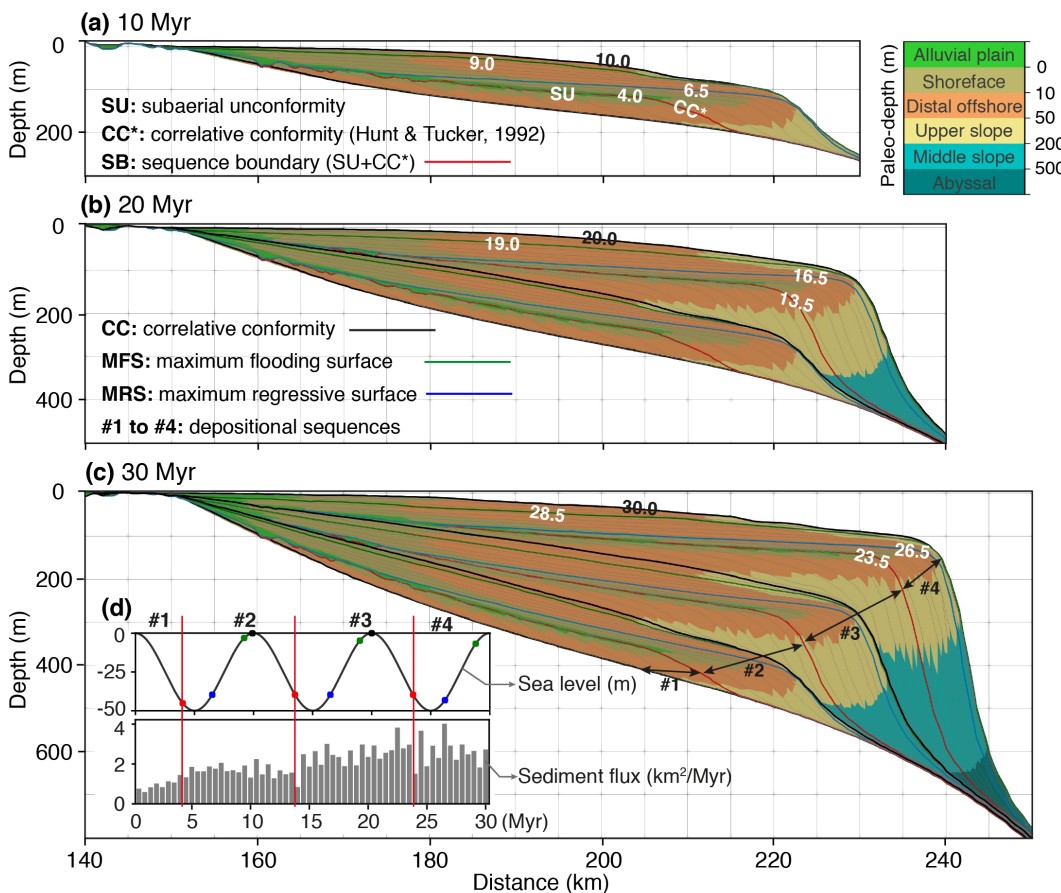

**Figure 5.** Temporal evolution of stratal stacking patterns at 10 Myr (a), at 20 Myr (b) and at 30 Myr (c). Lightgrey solid lines represent timelines at 0.5 Myr intervals. Colored solid lines with timing in Myr are identified stratigraphic surfaces. Stratal stacking patterns are coloured by paleo-depth used to represent different depositional environments. Four sequences #1 to #4 are defined. Inset in (c): Eustatic sea level curve and its rate of change. Colored dots indicate the timing of corresponding stratigraphic surfaces. The paleo-depth and topography shown in this figure are directly produced by our post-processing tool.

Myr (Fig. 5d). The stratigraphic units accumulated between 4.0 Myr and 13.5 Myr form a complete sequence (#2) bounded by two composite surfaces consisting of subaerial unconformities and correlative conformities.

Following the same criteria for identification of stratigraphic surfaces, a complete sequence (#3) is defined from 13.5 Myr to 23.5 Myr, with the maximum regressive surface formed at 16.5 Myr, the maximum flooding surface formed at 19.0 Myr and the correlative conformity formed at 20.0 Myr (Figs 5b and c). The sediment flux from 13.5 Myr to 23.5 Myr shows an increasing trend with large variations of up to 1.1 $\text{km}^2/\text{Myr}$, followed by a significantly drop from 2.8 $\text{km}^2/\text{Myr}$ to 1.5 $\text{km}^2/\text{Myr}$ at 23.5 Myr (Fig. 5d). From 23.5 Myr to 30 Myr, an incomplete sequence (#4) develops with a maximum regressive surface at

26.5 Myr and a maximum flooding surface at 28.5 Myr. The sediment flux during this time period shows two anomalous peaks at 24.0 Myr and 26.5 Myr (Fig. 5d).

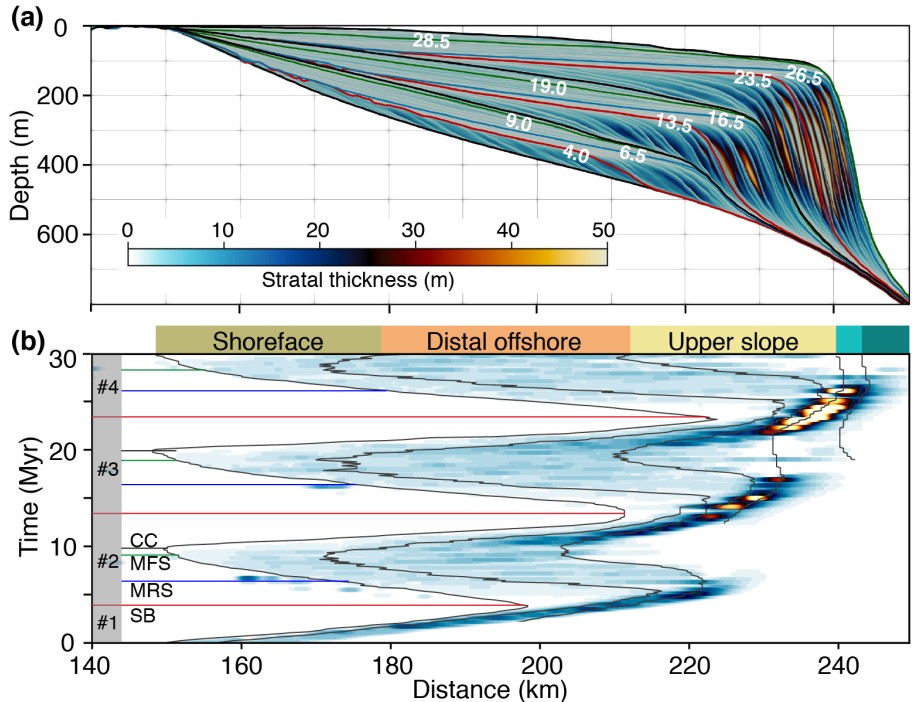

**Figure 6.** (a) Reconstructed stratal stacking pattern at 30 Myr showing the stratal thickness at 0.5 Myr intervals. (b) 3D Wheeler diagram or chronostratigraphy chart showing the horizontal distribution of depositional environment bounded by black solid lines and the accumulated sediment thickness through time along the extracted cross-section.

We also reconstructed the final stratal stacking pattern by computing stratal thickness in 100 Kyr increments (Fig. 6a), which constitutes the basis for a 3D Wheeler diagram (Fig. 6b). The 3D Wheeler diagram shows the horizontal distribution of depositional environment and the accumulated sediment thickness along the cross-section through time. Deposition hiatuses and condensed sections, which are essential to recognize bounding surfaces such as subaerial unconformities or sequence boundaries, are also denoted on the Wheeler diagram, as well as transgressive and flooding surfaces (Payton et al., 1977; Miall, 2004). The stratal thickness pattern shows three cycles, with thicker sediment accumulation during progradational and aggradational stratal stacking accompanied with low rate of eustatic sea level change, and thinner sediment accumulation during retrogradational stratal stacking accompanied with rising eustatic sea level. The stacking of sequences #1 to #4 shows an aggradational progradation pattern, with at least 10 km seaward progradation between successive cycles. Eustatic sea level fluctuations control sequence formations, as expected, and the effect of thermal subsidence and offshore sedimentation can be recognized when taking the stratigraphic package as a whole: The aggradational stacking reveals contributions of basement subsidence in creating accommodation, while the progradational stacking reveals the contributions of offshore sedimentation in decreas-

ing accommodation. When correlating the formation of stratigraphic surfaces with the horizontal migration of depositional packages, we find that the timing of maximum regressive surface agrees well with the onset of decreasing stratal thickness. Sequence boundaries correspond to the shift of shoreline from forestepping to backstepping, while correlative conformities correspond to the shift of shoreline from backstepping to forestepping.

## 4.2 Interpretation of depositional sequences

We now focus on the interpretation of the stratal architecture using both the trajectory analysis and accommodation succession methods.

### 4.2.1 Trajectory analysis

On the stratal stacking pattern extracted from the final output at 30 Myr, we manually pick the break-in-slope of the shelf-slope scale clinoforms as shelf-edge positions (magenta dots in Fig. 7a). Shoreline positions are difficult to pick on the cross-section because shoreline clinoforms are not well developed with this model setting. We therefore focus on the analysis of shelf-edge trajectory evolution. According to lateral and vertical shifts of the shelf edge through time, descending shelf-edge trajectory classes are identified from 0 to 5 Myr, 10 to 14 Myr, and 20 to 23 Myr; ascending shelf-edge trajectory classes are recognised from 5 to 6.5 Myr, 14 to 17 Myr and from 23 to 26.5 Myr, and transgressive shelf-edge trajectory classes are defined from 6.5 to 10 Myr, 17 to 20 Myr, and 26.5 to 30 Myr (Fig. 7a).

In addition to manually picking the shelf-edge trajectory on the final output, we developed post-processing tools to extract time-dependent shelf-edge and shoreline positions from successive outputs and interpret predicted depositional cycles accordingly. Figure 7b displays the extracted lateral and vertical migrations of the shelf-edge position and interpreted shelf-edge trajectory classes. The descending shelf-edge trajectory classes (DTC) is identified from 0 to 5.5 Myr, 10 to 13 Myr, and 20 to 22.5 Myr; the ascending shelf-edge trajectory class (ATC) is identified from 5.5 to 6.5 Myr, 13 to 16.5 Myr and from 22.5 to 25.5 Myr, and the transgressive shelf-edge trajectory class is identified from 6.5 to 10 Myr, 16.5 to 20 Myr, and 25.5 to 30 Myr (Fig. 7b). The shelf-edge trajectory classes identified through time generally resemble the ones identified manually from the final output, with differences in the timing of transition from one class to another by up to 1.5 Myr, which is greater than the temporal resolution (0.5 Myr) used to represent stratigraphic sequences (Figs 7a and b).

We now use the post-processing toolbox to identify changes in shoreline trajectories that are difficult to pick manually for this case. Figure 7c displays the automatically-detected lateral and vertical migrations of the shoreline position and interpreted shoreline trajectory classes accordingly. The descending regressive trajectory class (DRTC) is identified from 0 to 4 Myr, 10 to 13.5 Myr, and 20 to 23 Myr; the transgressive trajectory class (TTC) is identified from 5 and 10 Myr, 15 and 20 Myr, and 25 to 30 Myr (Fig. 7c). The transition from DRTC to TTC is concomitant with a depositional hiatus and the formation of an erosional surface, whereas the transition from TTC to DRTC is related to condensed stacking sections in the distal area induced by limited sediment supply (Fig. 5c). We note that between 4 and 5 Myr, 13.5 and 15 Myr, and 23 and 25 Myr, the shoreline migrates landward even though sea level is falling, that we call this trajectory type the "descending transgressive trajectory class" (DTTC) as this shoreline evolution is not described in the literature. In our models, the DTTC occurs because

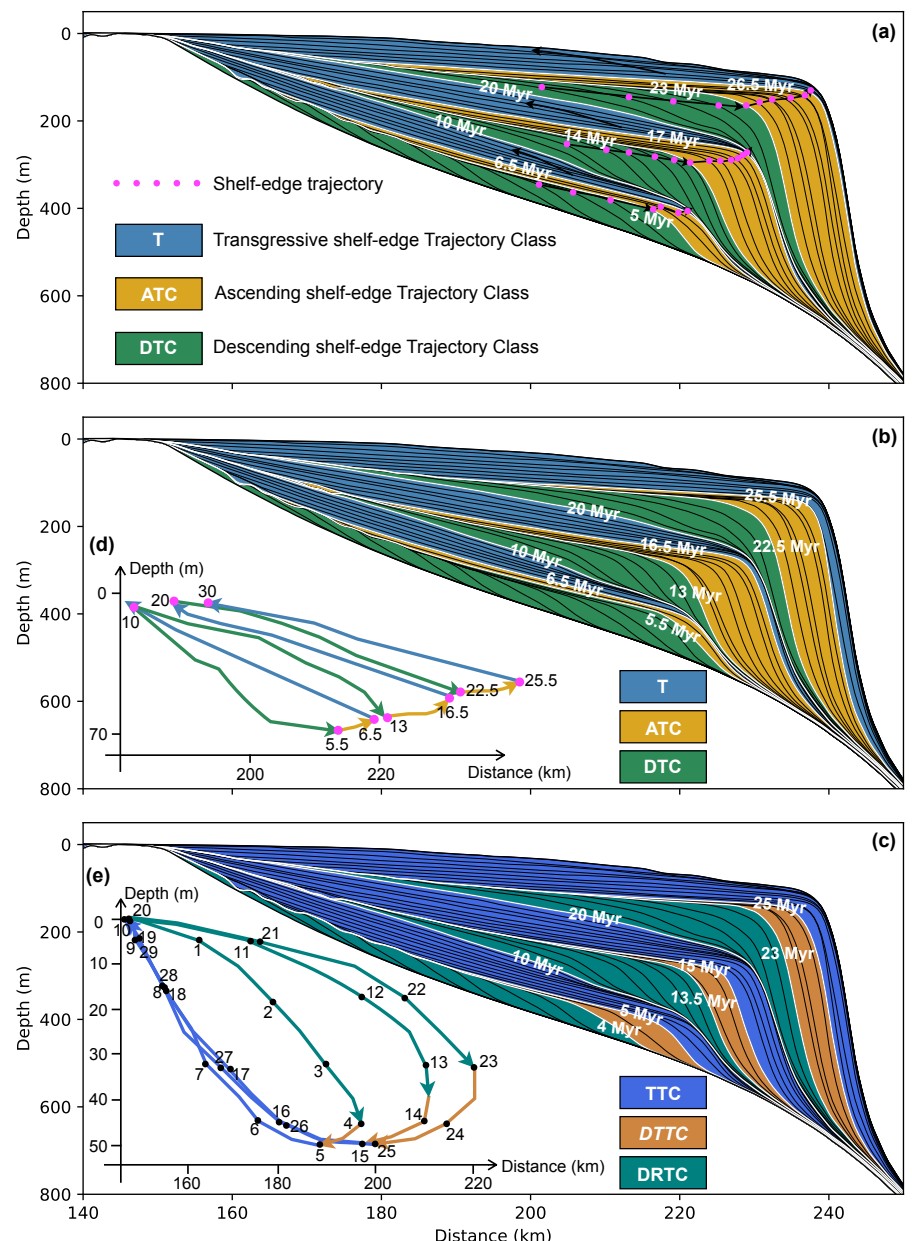

**Figure 7.** Interpretation of trajectory classes based on analysis of shoreline and shelf-edge trajectories (Helland-Hansen and Hampson, 2009). (a) Shelf-edge trajectory classes based on manually picking the shelf-edge trajectory (magenta dots) on the final output of stratal stacking pattern. (b) Automatically picking of shelf-edge trajectory classes based on calculated time-dependent shelf-edge positions in (d). (c) Automatically defined shoreline trajectory classes based on calculated time-dependent shoreline positions in (e). Time labels indicate the timing of each trajectory class formation. See Figure 2 for abbreviations.

the basement subsidence overrides the falling sea level and thus creates positive accommodation (Fig. 7e). This phenomenon is due to the model forcing conditions, and its identification directly linked to the time-dependent analysis of shoreline trajectory carried out here.

### 4.2.2 Accommodation succession analysis

Next, we apply the accommodation succession method to analyze the sequence formation in terms of changes in accommodation and sedimentation. Following the workflow proposed by Neal and Abreu (2016), we first manually marked stratal terminations (*i.e.* onlap, downlap, toplap) and offlap breaks on the simulated stratal stacking pattern (Step 1 in Fig. 8a). Here, the offlap break is defined as the shelf-edge rather than the shoreline as shoreline scale clinoforms do not develop with these model settings. Three stacking patterns and their bounding surfaces are then defined (Step 2 and Step 3 in Fig. 8a). For exam-

ple, toplaps and downlaps are observed in the first 3.5 Myr and correspond to progradational (P) stacking. The stratal geometry associated with P stacking is characterized by either erosion or by a lack of topset deposition and relatively thick clinoform front. Though strata keep downlapping, onlap terminations start replacing toplap terminations after 3.5 Myr, which reflects successive phases of progradation and aggradation. This depositional trend is defined as progradation to aggradation (PA) stacking, which is characterized by thin topset deposits and thick clinoform fronts. The unconformity formed between P and

PA stacking is interpreted as a sequence boundary (SB). Retrogradation (R) stacking corresponds to the onlapping of stratal deposits and landward shift of offlap break around 6.5 Myr. The thick topset deposit and condensed distal stacking characterize the stratal geometry deposited during this stage. Maximum regressive surfaces (MRS) are defined at the transition between PA and R stacking. At 10 Myr, the offlap break starts migrating seaward and downward. Toplap and downlap terminations are also observed after this time. The geometry of deposited strata also changes and is characterized by the formation of thicker

clinoform fronts. This stacking corresponds to the AP class (aggradation to progradation). Maximum trangressive surfaces (MTS) separate R stacking from AP stacking. At 13 Myr, onlap terminations are visible and the offlap break starts migrating upward, corresponding to the transition towards PA stacking just above the SB surface. Following the same criteria, successive stacking of PA, R and AP as well as bounding surfaces (SB, MTS & MRS) are defined on the cross-section (Fig. 8b). Finally, the interpreted R, AP, and PA stacking patterns are used to assess the changing history of $\delta A$ and $\delta S$ (Step 4 in Fig. 8b).

Instead of calculating $\delta A/\delta S$ as a ratio (Neal et al., 2016), we compute $\delta A - \delta S$ at the shoreline over time (Fig. 8d), because $\delta S$ can be equal to zero (Fig. 3). Under this approach, $\delta A - \delta S > 0$ corresponds to R stacking and is equivalent to $\delta A/\delta S > 1$; $\delta A - \delta S < 0$ and decreasing corresponds to APD stacking and is equivalent to $\delta A/\delta S < 1$ and decreasing; $\delta A - \delta S < 0$ and increasing corresponds to PA stacking and is equivalent to $\delta A/\delta S < 1$ and increasing. The stacking patterns can then be defined automatically (Fig. 8c), and are almost identical to the manually identified ones: differences in the timing of change of

stacking pattern are 0.5 Myr at most, which is equal to the temporal resolution (0.5 Myr) with which stratigraphic sequences are represented on Figures 7 and 8.

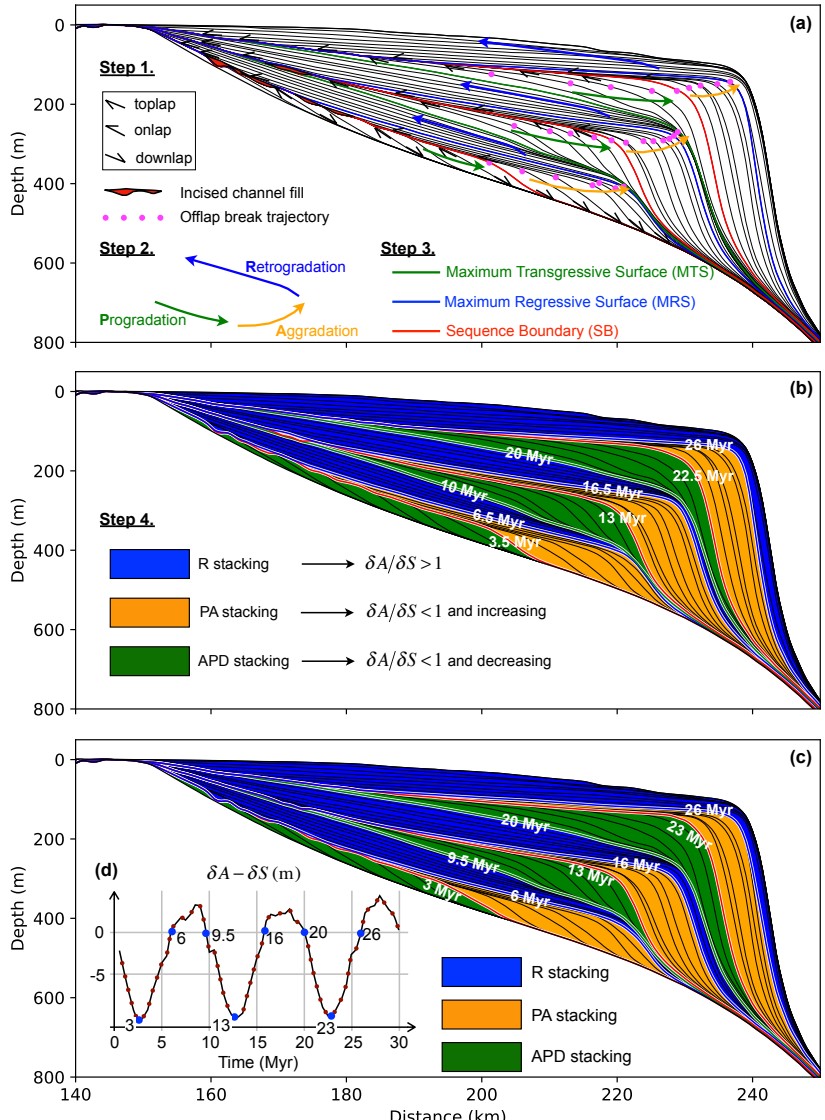

**Figure 8.** (a-b) Interpretation workflow based on the accommodation succession method (Neal et al., 2016). Step 1 includes marking stratal terminations (*i.e.* toplap, onlap and downlap represented using small arrows) and manually picking the break-in-slope as offlap break. The refilling of incised channels is shown in red, indicating erosional surfaces. Based on the marked stratal contacts, three stratal stacking trends (solid arrows) and three stratigraphic surfaces (colored solid lines) are then defined in Step 2 and Step 3. The three interpreted stacking patterns are filled with different colors, with their bounding times marked (Step 4). Each stacking pattern reflects the evolving ratio between rate of accommodation creation ($\delta A$) and rate of sediment supply ($\delta S$). (c) Automatically defined stacking patterns according to the calculated temporal evolution of $\delta A - \delta S$ (>0, <0 and decreasing, or <0 and increasing) (d).

## 5 Discussion

We compare and contrast the interpretations resulting from observations of temporal evolving stratal architecture (section **4.1**) with the interpretations resulting from the manual application of both trajectory analysis and accommodation succession methods (section **4.2.1**) and from quantitative analysis of these two methods using our post-processing tools (section **4.2.2**).

The manual application of shelf-edge trajectory analysis presents reliable interpretations of transgressive stratigraphic units (T) but displays notable discrepancies in separating descending stratigraphic units (DTC) from ascending stratigraphic units (ATC) especially during in the early stages of deposition (Fig. 7a). Note that here we only discuss discrepancies beyond 0.5 Myr which is the time interval of reconstructing stratal stacking patterns in section **4.2**. The imposed thermal subsidence modifies the position of the shelf-edge positions after its formation (Fig. 9): the shelf-edge trajectory appears to rise between 3.5 Myr and 6.5 Myr on the snapshot at 10 Myr (Fig. 9a), whereas it appears to fall between 3.5 Myr and 5 Myr on the snapshot at 20 Myr (Fig. 9b), because of ongoing thermal subsidence between 10 Myr and 20 Myr. As a consequence, identifying strata on the final output artificially extends the duration of descending trajectory class (Figs 7a and 9b). This should be kept in mind when identifying shelf-edge trajectories on seismic profiles, which are by nature a snapshot of the evolution of a basin. Constraining time-dependent processes such as thermal subsidence from sedimentary packages would make it would be useful to correct shoreline/shelf-edge trajectories for these processes.

The quantitative shelf-edge trajectory analysis reveals discrepancies in defining ascending trajectory classes at 5.5 Myr, 22.5 Myr and 25.5 Myr (Fig. 7b), as extracting the position of the shelf-edge is more uncertain for steep strata. The tool we have developed can be applied to actual sections as long as seismic timelines are accessible. Again, we emphasize that additional constraints from observation of stratal geometry would improve the interpretations of stratigraphic units. In terms of quantitative shoreline trajectory analysis, the distinction between the descending transgressive trajectory class and the transgressive trajectory class is controlled by the eustatic sea level fluctuations (Fig. 7c). Since the shoreline and the shelf-edge represent the break-in-slope of clinoforms at different scales (Patruno et al., 2015, Fig. 2), it is not appropriate to apply shoreline trajectory analysis to shelf-slope clinoforms. However, the numerical tools we provided to extract time-dependent shoreline positions based on a given sea-level forcing would also be useful and applicable to short-term numerical and analogue experiments (Martin et al., 2009; Granjeon et al., 2014).

The stratigraphic interpretations from the accommodation succession method indicate that there is no significant difference between analysing the final output and time-dependent outputs (Figs 8b and 5). Therefore, the analysis of the presented model is more robust with the accommodation succession method than with the trajectory analysis method.

The quantitative accommodation succession analysis also shows largely consistent interpretations (Figs 8c and 5), except for a 1-Myr discrepancy in demarcating aggradation to progradation to degradation stacking (APD) and progradation to aggradation stacking (PA) at 3 Myr. We find that the $\delta A - \delta S$ curve (Fig. 8d) presents trends similar to the rate of eustatic sea level change (Fig. 4b). This suggests that the evolution of $\delta A - \delta S$ is a proxy for the derivative of sea level change with respect to time, rather than a direct proxy for sea level change. However, difference exists between the $\delta A - \delta S$ curve and the rate of sea level change curve: the $\delta A - \delta S$ curve shows asymmetrical fluctuations with larger amplitude below zero than above zero,

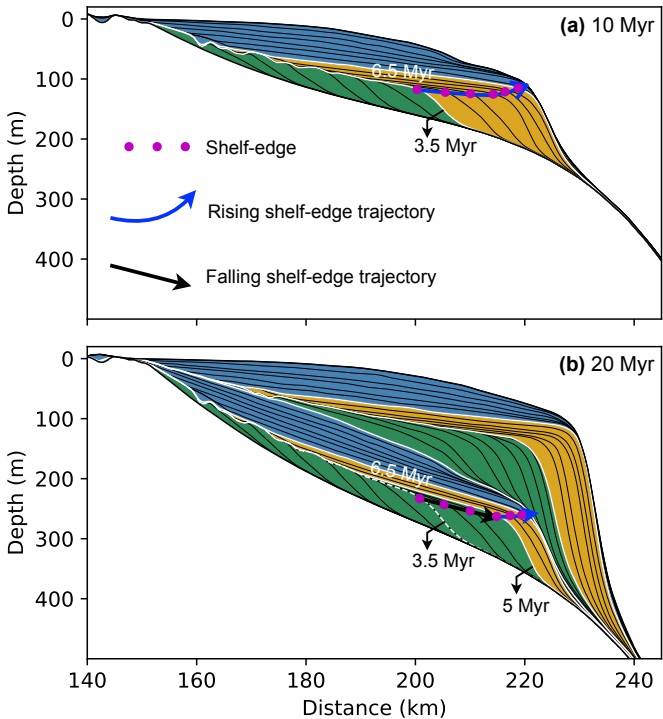

**Figure 9.** Stratal stacking pattern extracted at 10 Myr (a) and 20 Myr (b). Between 3.5 Myr and 5 Myr, the shelf-edge trajectory changes from rising at 10 Myr to falling at 20 Myr, as a result of basement subsidence. The descending trajectory class thus extends to 5 Myr.

which is attributed to the sediment supply. Discrepancies of <0.5 Myr are observed between the $\delta A - \delta S$ curve and the rate of eustatic sea level change curve, which are likely to be related to the temporal resolution (0.5 Myr) used to compute $\delta A - \delta S$.

A common issue when calculating the ratio $\delta A/\delta S$ is the lack of unique approaches and common dimensional units to define these two metrics (Muto and Steel, 2000; Burgess, 2016). Both metrics represent changes in volume over a specific time interval and thus should be defined in m³/yr. However, difficulties in delineating the potential space for sediment deposition require the use of proxies to quantify accommodation. Although the sedimentation rate is often used as a proxy for $\delta S$ (Poag and Sevon, 1989; Galloway and Williams, 1991; Liu and Galloway, 1997; Galloway, 2001), it only provides information about the deposition at a single location and does not reflect the spatial distribution of sedimentation (Petter et al., 2013). Furthermore, the distribution of sediment deposition is not determined solely by sediment supply, but is a combined result of the distance to sediment source, basin physiography and sediment transport efficiency (Posamentier et al., 1992; Posamentier and Allen, 1993). Recently, new methods have been proposed to improve the quantification of $\delta S$. Petter et al. (2013) proposed an approach that directly reconstructs sediment paleo-fluxes from stratigraphic records. Ainsworth et al. (2018) used a technique termed "TSF analysis" in which parasequence thickness (T) is used as proxy for accommodation at the time of deposition while parasequence sandstone fraction (SF) as proxy for sediment supply. Our work could be used to integrate and test these new quantitative interpretations based on the evolution of accommodation and sedimentation in a stratigraphic modelling

framework. The quantification of $\delta A$ and $\delta S$ presented here could be extended in future work to investigate the interplay between accommodation change and sediment supply in 3D.

Our source-to-sink numerical scheme generates 3D landscape evolution and stratigraphic development, though only 2D stratigraphic architectures are extracted along dip-oriented cross-sections. To evaluate the lateral variations in sequence forma-
tion potentially induced by sediment diffusion in offshore environment and lateral migrations of river mouth, we extract five dip-oriented cross-sections (CS1 to CS5) that are parallel to each other and two along-strike cross-sections (Fig. 10). Cross-section CS3 is the same one as presented in the Result section. The timing of key stratigraphic surfaces on CS1 to CS5 is showed in Table 1, with differences varying from 0.5 Myr to 1.5 Myr. The timing of sequence boundaries shows the most variations, compared to other stratigraphic surfaces. The timings of correlative conformities of sequence #2 (CC1) and sequence
#3 (CC2) are consistent on cross-sections CS1 to CS5 and correspond to the onset of eustatic sea level fall. The stacking of depositional environment along strike shows increasing variations towards the basin. For the presented case there is overall little variation in stratigraphic sequences across strike and along strike, which is expected from the model setup. The presented tools can be used for the 3D stratigraphic analysis of more complex cases.

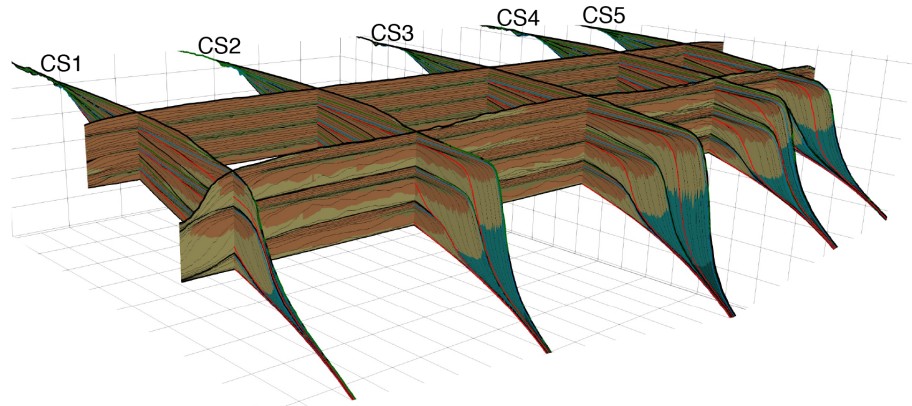

**Figure 10.** Stratal stacking patterns on five dip-oriented cross-sections (CS1 to CS5) and two along-strike cross-sections. Key stratigraphic surfaces on CS1 to CS5 are identified and colored accordingly.

We have explored stratigraphic development in a source-to-sink context in which the dynamic sediment supply to the passive
continental margin depends on climatic and tectonic evolution in the source area. Therefore, the physiographic evolution of the upstream region controls the depositional patterns in the sink area together with accommodation change (Ruetenik et al., 2016; Li et al., 2018). Most stratigraphic forward models (SFMs) focus on simulating sediment transport and deposition in the sink area, which limits the interpretation of the effect climatic and tectonic evolution on the stratigraphic record. In our framework, sediment transport and supply to the margin is dynamically related to autogenic and allogenic processes. As an
example, though forced with uniform rainfall pattern in the source region, the rate of sediment supply to the sink area fluctuates with time (Fig. S1). This highlights the complex relationships between erosional signals and the preservation of a depositional record (Van Heijst et al., 2001; Kim et al., 2006; Kim, 2009). The source-to-sink numerical scheme used in *pyBadlands* makes

**Table 1.** Timing of stratigraphic surfaces on CS1 to CS5 (Myr)

| | #1 | #2 | | | | #3 | | | | #4 | |
|---|---|---|---|---|---|---|---|---|---|---|---|
| | SB1 | MRS1 | MFS1 | CC1 | SB2 | MRS2 | MFS2 | CC2 | SB3 | MRS3 | MFS3 |
| CS1 | 3.5 | 6.5 | 9.0 | 10.0 | 13.0 | 16.5 | 18.5 | 20.0 | 23.5 | 26.5 | 28.5 |
| CS2 | 3.5 | 6.5 | 8.5 | 10.0 | 12.0 | 17.0 | 19.0 | 20.0 | 23.0 | 26.5 | 28.0 |
| **CS3** | 3.5 | 6.5 | 9.0 | 10.0 | 13.5 | 16.5 | 19.0 | 20.0 | 23.5 | 26.5 | 28.5 |
| CS4 | 3.5 | 6.5 | 9.0 | 10.0 | 13.5 | 17.0 | 18.5 | 20.0 | 23.0 | 26.5 | 28.5 |
| CS5 | 3.0 | 5.5 | 9.0 | 10.0 | 13.0 | 17.0 | 19.0 | 20.0 | 23.0 | 27.0 | 29.0 |

it possible to explore important questions for the future of sequence stratigraphy, such as the role of sediment supply variations in the generation of stratigraphic sequences at different temporal scales (Burgess, 2016), and the importance of allogenic and autogenic processes in the formation and evolution of stratal record (Paola, 2000; Paola et al., 2009).

We modelled the formation of stratigraphic sequence over 30 Myr, which represents second to third order stratigraphic cycles
(Vail et al., 1977b). Over this temporal scale, long-term eustatic sea level changes and dynamic uplift or subsidence induced by tectonics or deep Earth processes (e.g. mantle flow driven dynamic topography) might drive deposition (Burgess and Gurnis, 1995; Burgess et al., 1997), moderated by higher-frequency fluctuations in climate and sea level. The long-term stratigraphic record along continental passive margins thus potentially contains important constrains on the evolution of the structure of the deep Earth (Mountain et al., 2007; Braun, 2010; Flament et al., 2013; Salles et al., 2017). The framework we have introduced in
this study integrates both long-term surface processes and stratigraphic modelling and can be used to quantitatively investigate the influences of long-term tectonics and deep Earth dynamics on stratal geometries and depositional patterns evolution as well as their feedback mechanisms (Jordan and Flemings, 1991; van der Beek et al., 1995; Rouby et al., 2013). Note that the tools we have introduced here are not specific to any temporal or spatial scale, and can also be used for short-term stratigraphic modelling.

**6 Conclusions**

We used *pyBadlands* to model sediment erosion, transport and deposition from source to sink, and to investigate the formation of stratigraphic sequences on a passive continental margin in response to long-term sea level change, thermal subsidence, and dynamic sediment supply. We analysed predicted stratigraphic architecture based on observations of shelf-edge or offlap break trajectory, stratal terminations and stratal geometries, following the workflow of two stratigraphic interpretation approaches:
the trajectory analysis and the accommodation succession method. We introduced a set of post-processing tools to extract the temporal evolution of shoreline, shelf-edge, rate of accommodation change ($\delta A$) and sedimentation $\delta S$, based on which automatic interpretations can be obtained. Our results suggest that the stacking patterns defined with the accommodation succession method provide more robust reconstructions of the changing history of accommodation and sedimentation than the

trajectory analysis method, because the interpretation of sequences with the trajectory analysis depends on time. In contrast, the accommodation succession method is not affected by the time-dependence of processes controlling the evolution of deposition. As a consequence, seismic data should be backstripped before stratigraphic sequences are intrepreted using the trajectory analysis method. Our work presents an integrated workflow that can be used to generate 2D and 3D stratal architectures on
basin margins and to interpret stratigraphic sequences produced by large-scale and long-term numerical experiments.

*Code availability.* *pyBadlands* is an open-source package distributed under GNU GPLv3 license. The source code is available on GitHub (http://github.com/pyBadlands-model). The easiest way to use the code is via our Docker container (pyBadlands-serial) which is shipped with the complete list of dependencies, the model companion and a series of examples. The code, inputs and post-processing functions used in this study are available on GitHub (https://github.com/XuesongDing/GMD-models).

*Author contributions.* Xuesong Ding designed the experiments and analysed the outputs with all co-authors. Tristan Salles developed the model code. Xuesong Ding prepared the manuscript with contributions from all co-authors.

*Competing interests.* The authors declare that they have no conflict of interest.

*Acknowledgements.* XD, TS and PR were supported by ARC grant IH130200012 and NF was supported by ARC grant DE160101020. We also acknowledge the support from International Association for Mathematical Geosciences (IAMG) grant. This research was undertaken
with the assistance of resources and services from the National Computational Infrastructure (NCI), which is supported by the Australian Government.

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
