# Peer review of "Quantitative stratigraphic analysis in a source-to-sink numerical framework"

_Geoscientific Model Development, 2018_

## Referee Comment (RC1) · Sylvester (Referee) · 25 Dec 2018

General comments

This paper describes the results of a numerical experiment focusing on erosion and sedimentation along a continental margin and the stratigraphic analysis of the model deposits. Unlike most previous modeling studies, the erosional evolution of the sediment source area is coupled with sedimentation along the coast. I think it is overall well-written, nicely illustrated, to the point, and, most importantly, an interesting and valuable contribution to the modeling and sequence stratigraphic literature. It illustrates well the power and elegance of the pyBadlands modeling package. In addition, it is hard to overestimate the value of having easy access to both the modeling software
and the scripts that were used to generate the model in the paper.

Specific comments

Although I think the paper is a valuable contribution as it is, there are a number of points that could be addressed to make it more comprehensive:

1. Although the authors convincingly show how the trajectory analysis and accommodation succession approaches can be applied to the model results, both manually and in an automated way, and they conclude that the accommodation succession method is more robust, they do not spell out suggestions for practitioners of stratigraphic interpretation. Is a manual approach good / reliable enough? Is it possible to automate the interpretation of actual sections, not just model sections? Should the idea of using dA/dS (as opposed to, let's say, dA-dS) be entirely abandoned? Is it acceptable to talk about dA and dS without specifying what they exactly mean and quantifying them? There seems to be a good opportunity to expand on these issues in the Discussion section.

2. In many, maybe most cases the purpose of stratigraphic interpretation is not just subdivision into meaningful units, but a reconstruction of different forcing parameters / signals. How do the models and analysis shown here perform in this regard? E.g., can the dA-dS curve (Figure 7d) be used as a proxy for sea level? What is the significance of the ~2 Ma phase shift between the two? This could be the subject of another paper, but it is probably worth exploring it briefly here as well.

3. The analysis assumes that a single cross section through the model is representative of the whole model / continental margin. The model setup makes it likely that this is indeed the case, but it would be useful to show how similar / dissimilar are other cross sections. Would the analysis of s different section come up with a very similar result? What if there are a significant number of delta lobe avulsions? Again, I realize that a detailed investigation of this could form the subject of another paper, but this question should be addressed. In its current form, this study seems to wholeheart-

edly encourage sequence stratigraphic interpretation based on single dip sections; yet many real-life deltas are highly three-dimensional and single cross sections do not record the history of the entire system.

Technical corrections

Page 1, lines 2-4 (and throughout the paper): I am not sure that it is worth reiterating the idea of dA/dS as a key parameter in stratigraphy. You end up using dA-dS anyway; and dA is defined here as the rate of relative sea level change, dS as sedimentation rate. Why not refer to the actual parameters used?

Page 2, line 8 – 'tectonics' instead of 'tectonic'

Page 2, line 9 – cut 'to stratigraphic interpretations'

Page 3, line 2 – 'automate' instead of 'automatise'

Page 3, line 6 – 'interpretation' instead of 'interpretations'

Page 3, line 12 – 'designed the trajectory analysis technique'

Page 4, line 3 – 'First,' instead of 'Firstly'

Page 4, line 8 – 'the topographic contour that corresponds to sea level'

Page 4, line 9 – 'a critical slope of 0.025 degrees.'

Page 6, lines 7-9 – probably should mention that the model setup focuses on sea level changes, as both climate (precipitation) and subsidence patterns are kept constant. Sediment input increases through time, but it does not vary periodically as sea level does.

Page 6, line 10 – 'sequence development' instead of 'sequences development'.

Page 9, figure 5 – what is the horizontal scale in (b)? Tickmarks do not match those in (c). Stratigraphic columns in (d) do not seem to match the ones in (a).

[Figure]

Page 9, line 2 – 'three stratigraphic cycles' (?) instead of 'three cyclical vertical stacking'

Page 9, line 3 – 'apparent in' instead of 'apparent on'

Page 9, lines 4-5 – cut 'the vertical stacking pattern'

Page 10, line 1 – 'Interpretation' instead of 'Interpretations'

Page 10, line 2 – 'both the trajectory' instead of 'both trajectory'

Page 10, line 7 – 'difficult to pick' instead of 'difficult to be picked'

Page 10, line 9 – 'According to lateral and vertical shifts of the shelf edge through time,'

Page 10, line 29 – 'We call this trajectory type the "descending. . .'

Page 11, figure 6 – is the first segment of the first ATC trajectory really ascending in (d)? Seems descending to me.

Page 12, line 2 – 'Next, we. . .' instead of 'We then. . .'

Page 12, line 5 – '. . .clinoforms do not develop with these model settings.' instead of 'clinoforms are not well generated in this model setting.'

Page 12, line 7 – 'progradational (P)' instead of 'progradation (P)'

Page 14, line 3 – 'from the final output' instead of 'from final output'

---

## Referee Comment (RC2) · Neal (Referee) · 31 Jan 2019

General Comments: "Quantitative stratigraphic analysis in a source-to-sink numerical framework" by Xuesong Ding et al. is a clearly written and thoughtful submission that can be a strong contribution after significant technical clarification is included. It might also be better titled, considering the content is dominated by a comparative analysis of alternate sequence stratigraphic interpretation methods using manual and automated means to compare the fit of results with pyBadlands Stratigraphic Forward Model (SFM) input and output. The approach used is novel, applying different interpretation techniques on the output of a SFM and comparing the results of each technique against time-dependent SFM inputs and outputs. Unfortunately, there are flaws in the analysis that stem from a blurring of observations that are the foundation of interpretation

methods and the forcing mechanism inferred to drive them. • Firstly, a $\beta$-factor of 1 to 2.5 over 150 km produces a subsidence profile which increases so much toward the basin that 10 million year duration, 50m "eustasy" cycles don't produce basinward shifts of facies (depositional sequence boundaries) resulting from negative shelfal accommodation that is a key factor to interpretation with either shoreline trajectory (ST; Helland-Hansen and others '94-'09) or accommodation succession (AS; Neal and others '09-'16) methods. • Application of ST method is disadvantaged as presented because the SFM produces a trajectory the authors had to invent ("descending transgressive trajectory class" or DTTC) in order to fit geometries with known sea-level conditions. This is a limitation to methods that are explicitly linked to sea-level change. • The AS method explicitly avoids sea-level requirements and focuses on stratal terminations at key surfaces that bound different stacking patterns. This method allows interpretation to adjust to dipping strata that was initially horizontal (clinoform topsets – coastal plain aggradation) • ST method builds from the assumption of trajectory from horizontal, so differentiating relative to AS is artificial (a function of forcing it to fit the sea level curve). THIS is the actual insight from Ding et al.'s paper – apply ST or AS methods but do not force them to fit a sea level curve. We don't observe sea level in stratigraphy, we infer it. We observe stratal terminations, shoreline trajectories, vertical and lateral stacking of facies associations, and key bounding surfaces that record significant changes in these observations. • The erosion feature of pyBadlands produces interesting 2D truncation geometries updip (but this was not demonstrated in the Wheeler diagram (fig. 5c) and might produce more interesting relations in shoreline trajectory if $\beta$-factor were reduced. For scaling comparison, I suggest you refer to the physical flume model and resulting interpretation published in Martin et al. 2009 (Martin, J., Abreu, V., Neal, J. Sheets, B. 2009. Sequence stratigraphy of experimental strata under known conditions of differential subsidence and variable base level. AAPG Bulletin, 93, 503–533.) • In summary, there are ways this experiment could be run that would make a better comparison of interpretation methods or the paper could more directly highlight shortcomings of interpretation methods that are explicitly

linked to sea-level change. The approach in Ding et al. is innovative for using SFM to volumetrically quantify $\delta A/\delta S$ or ($\delta A$ - $\delta S$ if you wish) and I encourage the authors clarify their purpose (change the model or change the conclusions and application) so this good work is more on target.

―――――――――――――――――

---

## Author Comment (AC1) · 24 Apr 2019

(General comments) Comment 1: This paper describes the results of a numerical experiment focusing on erosion and sedimentation along a continental margin and the stratigraphic analysis of the model deposits. Unlike most previous modelling studies, the erosional evolution of the sediment source area is coupled with sedimentation along the coast. I think it is overall well-written, nicely illustrated, to the point, and, most importantly, an interesting and valuable contribution to the modelling and sequence stratigraphic literature. It illustrates well the power and elegance of the pyBadlands modelling package. In addition, it is hard to overestimate the value of having easy access to both the modelling software and the scripts that were used to generate the model in the paper.

Response: We acknowledge the reviewer's appreciation of the importance and value of this work.

(Specific comments) Comment 2: Although the authors convincingly show how the trajectory analysis and accommodation succession approaches can be applied to the model results, both manually and in an automated way, and they conclude that the accommodation succession method is more robust, they do not spell out suggestions for practitioners of stratigraphic interpretation. Is a manual approach good / reliable enough? Is it possible to automate the interpretation of actual sections, not just model sections? Should the idea of using dA/dS (as opposed to, let's say, dA-dS) be entirely abandoned? Is it acceptable to talk about dA and dS without specifying what they exactly mean and quantifying them? There seems to be a good opportunity to expand on these issues in the Discussion section.

Response: Thank you for raising these insightful questions. In the Discussion, we added detailed comparisons of stratigraphic interpretations resulting from different approaches and then proposed suggestions for practical applications (Page 15, from line 7; Page 16, lines 1-2). The manual application of the accommodation succession method provides reliable interpretations, while the trajectory analysis depends on time-dependent processes such as thermal subsidence. It is possible to automate the interpretation of actual sections using the shelf-edge trajectory analysis. Again, we showed that corrections of time-dependent processes would be required beforehand. Also, constraints from stratal geometry would be useful to correct possible modifications of shoreline/shelf-edge trajectories by contributing processes. We did not intend to replace dA/dS with dA-dS. We used dA-dS because in our calculation dS could be zero. We clearly stated the meaning of both dA and dS. Due to difficulties in quantifying the "true" dA and dS, we used relative sea level change and sedimentation rate at the time-dependent shoreline as proxies for dA and dS to quantify the competing between dA and dS through time.

Comment 3: In many, maybe most cases the purpose of stratigraphic interpretation

is not just subdivision into meaningful units, but a reconstruction of different forcing parameters / signals. How do the models and analysis shown here perform in this regard? E.g., can the dA-dS curve (Figure 7d) be used as a proxy for sea level? What is the significance of the ~2 Ma phase shift between the two? This could be the subject of another paper, but it is probably worth exploring it briefly here as well.

Response: Applying the objective accommodation succession method makes it possible to reconstruct the evolution of dA/dS. We correlated the timing and development of stratigraphic units with eustatic sea level changes and sediment supply, and found that the dA-dS curve (Figure 8d) has similar changing trends to the rate of eustatic sea level change (Figure 4b). This suggests that the evolution of dA-dS is a proxy for the derivative of sea level change with respect to time, rather than a direct proxy for sea level change. Discrepancies of <0.5 Myr are observed between the dA-dS curve and the rate of eustatic sea level change curve, which are likely to be related to the temporal resolution (=Âă0.5ÂăMyr) used to compute dA-dS.

Comment 4: The analysis assumes that a single cross section through the model is representative of the whole model / continental margin. The model setup makes it likely that this is indeed the case, but it would be useful to show how similar / dissimilar are other cross sections. Would the analysis of s different section come up with a very similar result? What if there are a significant number of delta lobe avulsions? Again, I realize that a detailed investigation of this could form the subject of another paper, but this question should be addressed. In its current form, this study seems to wholeheartedly encourage sequence stratigraphic interpretation based on single dip sections; yet many real-life deltas are highly three-dimensional and single cross sections do not record the history of the entire system.

Response: Thank you for raising this important point. In the Discussion, we added a new figure (Page 17, Figure 10) that shows five dip-oriented cross-sections and two along-strike cross-sections. The accumulation of depositional environments (Figure 10a) and stratal thickness (Figure 10b) are reconstructed on these cross-sections. We

observed notable differences in the stratal thickness along-strike, while the formation of stratigraphic surfaces is laterally consistent.

(Technical corrections) Comment 5: Page 1, lines 2-4 (and throughout the paper): I am not sure that it is worth reiterating the idea of dA/dS as a key parameter in stratigraphy. You end up using dA-dS anyway; and dA is defined here as the rate of relative sea level change, dS as sedimentation rate. Why not refer to the actual parameters used?

Response: dA/dS is the most widely-used way to analyse the competition between the rate of change of accommodation creation (dA) and the rate of change of sediment supply (dS). As dS can be zero at the shoreline in our model predictions, we use dA-dS instead. The concept of dA/dS is useful, although it remains challenging to quantify this indicator. Future work could comprehensively explore the interplay between accommodation and sediment supply, especially in 3D depositional systems.

Comment 6: Page 2, line 8 – 'tectonics' instead of 'tectonic'

Response: We changed 'tectonic' to 'tectonics' (Page 2, line 8).

Comment 7: Page 2, line 9 – cut 'to stratigraphic interpretations'

Response: We rephrased the sentence (Page 2, lines 9-11).

Comment 8: Page 3, line 2 – 'automate' instead of 'automatise'

Response: We rephrased the sentence (Page 3, lines 12-14).

Comment 9: Page 3, line 6 – 'interpretation' instead of 'interpretations'

Response: We reorganized the configuration of sections by combing the previous section 2 and section 3 into the current section 2 - 'Quantitative stratigraphic analysis in pyBadlands' (Page 3, line 18). The figures within the previous sections 2 and 3 were rearranged accordingly.

Comment 10: Page 3, line 12 – 'designed the trajectory analysis technique'

Response: We rephrased the sentence (Page 3, lines 27-28).

Comment 11: Page 4, line 3 – 'First,' instead of 'Firstly'

Response: We changed 'Firstly' to 'First' (Page 3, line 26).

Comment 12: Page 4, line 8 – 'the topographic contour that corresponds to sea level'

Response: We changed the text 'the topographic contour equals to sea level' to 'the topographic contour that corresponds to sea level' (Page 5, line 7).

Comment 13: Page 4, line 9 – 'a critical slope of 0.025 degrees.'

Response: We modified the text 'a critical slope 0.025 degree' to 'a critical slope of 0.025 degrees' (Page 5, lines 7-8).

Comment 14: Page 6, lines 7-9 – probably should mention that the model setup focuses on sea level changes, as both climate (precipitation) and subsidence patterns are kept constant. Sediment input increases through time, but it does not vary periodically as sea level does.

Response: We modified that paragraph to 'Considering that this study focuses on long-term stratigraphic evolution related with sea level changes, both climate and subsidence patterns are kept constant. Climate is assumed to be directly related to precipitation with a spatially and temporally uniform precipitation rate of 2.0 m/yr over 30 Myr. Sediment input varies through time, depending on the dynamic evolution of source area.' (Page 7, lines 5-8).

Comment 15: Page 6, line 10 – 'sequence development' instead of 'sequences development'.

Response: We changed 'sequences development' to 'sequence development' (Page 7, line 9).

Comment 16: Page 9, figure 5 – what is the horizontal scale in (b)? Tickmarks do not

match those in (c). Stratigraphic columns in (d) do not seem to match the ones in (a).

Response: We modified Figure 5 based on this comment (Page 9, Figure 5). Furthermore, we split the original Figure 5 into two figures (Figure 5 and Figure 6). In Figure 5 (Page 9), we presented snapshots of stratal stacking patterns at 10 Myr, 20 Myr and 30 Myr. The Wheeler diagram was moved to Figure 6 (Page 10), and was rebuilt to be 3D by adding the information of stratal thickness (Figure 6b). In Figure 6, we also showed the stratal thickness within the stratal stacking pattern (Figure 6a).

Comment 17: Page 9, line 2 – 'three stratigraphic cycles' (?) instead of 'three cyclical vertical stacking' Comment 18: Page 9, line 3 – 'apparent in' instead of 'apparent on'

Comment 19: Page 9, lines 4-5 – cut 'the vertical stacking pattern'

Response: We removed the result of vertical stacking patterns.

Comment 20: Page 10, line 1 – 'Interpretation' instead of 'Interpretations'

Response: We changed 'interpretations' to 'interpretation' (Page 11, line 9).

Comment 21: Page 10, line 2 – 'both the trajectory' instead of 'both trajectory'

Response: We changed 'both trajectory' to 'both the trajectory' (Page 11, line 10).

Comment 22: Page 10, line 7 – 'difficult to pick' instead of 'difficult to be picked'

Response: We modified 'difficult to be picked' to 'difficult to pick' (Page 11, line 14).

Comment 23: Page 10, line 9 – 'According to lateral and vertical shifts of the shelf edge through time,'

Response: We modified the text 'According to its lateral and vertical shifts through time' to 'According to lateral and vertical shifts of the shelf edge through time' (Page 11, line 16).

Comment 24: Page 10, line 29 – 'We call this trajectory type the "descending. . .'

Response: We changed 'name' to 'call this trajectory type' (Page 13, line 3).

Comment 25: Page 11, figure 6 – is the first segment of the first ATC trajectory really ascending in (d)? Seems descending to me.

Response: Thank you for pointing this out. We re-examined the shelf-edge trajectory and agreed that the shelf-edge is descending from 3.5-5.5 Myr, and therefore expanded the subdivision of DTC from 0-3.5 Myr to 0-5.5 Myr (Page 12, Figure 7b, 7d).

Comment 26: Page 12, line 2 – 'Next, we. . .' instead of 'We then. . .'

Response: We changed 'We then' to 'Next, we' (Page 13, line 9).

Comment 27: Page 12, line 5 – '. . .clinoforms do not develop with these model settings.' instead of 'clinoforms are not well generated in this model setting.'

Response: We changed 'clinoforms are not well generated in this model setting' to 'clinoforms do not develop with these model settings' (Page 13, lines 12-13).

Comment 28: Page 12, line 7 – 'progradational (P)' instead of 'progradation (P)'

Response: We changed 'progradation (P)' to 'progradational (P)' (Page 13, line 14).

Comment 29: Page 14, line 3 – 'from the final output' instead of 'from final output'

Response: We rephrased the sentence (Page 15, lines 13-14).
* * *
**(a)** 10 Myr

Depth (m)

9.2

9.9

SU

4.0

6.5
CC*

**SU:** subaerial unconformity
**CC*:** correlative conformity (Hunt & Tucker, 1992)
**SB:** sequence boundary (SU+CC*) ———

Paleo-depth (m)

| | |
|---|---|
| Alluvial plain | 0 |
| Shoreface | 10 |
| Distal offshore | 50 |
| Upper slope | 200 |
| Middle slope | 500 |
| Abyssal | |

**(b)** 20 Myr

Depth (m)

19.0

20.0

16.5

13.5

**CC:** correlative conformity ———
**MFS:** maximum flooding surface ———
**MRS:** maximum regressive surface ———
**#1 to #4:** depositional sequences

**(c)** 30 Myr

Depth (m)

28.8

30.0

23.5   26.2

**4**

**3**

**2**

**1**

**1    #2       #3    #4**

Sea level (m)

0
-25
-50

15
0
-15

Rate of sea level
change (m/Myr)

0   5   10  15  20  25  30 (Myr)

140        160        180        200        220        240

Distance (km)

**Fig. 1.** Figure 5

**(a)**

Stratal thickness (m)

**(b)**

Shoreface    Distal offshore    Upper slope

MFS
MRS
SB
CC
MFS
MRS
SB
CC
MFS
MRS
SB

**Fig. 2.** Figure 6

[Figure]

**Fig. 3.** Figure 7

**(a)**

[Figure]

**(b)**

[Figure]

**Fig. 4.** Figure 10

---

## Author Comment (AC2) · 24 Apr 2019

(General comments) Comment 1-1: "Quantitative stratigraphic analysis in a source-to-sink numerical framework" by Xuesong Ding et al. is a clearly written and thoughtful submission that can be a strong contribution after significant technical clarification is included. It might also be better titled, considering the content is dominated by a comparative analysis of alternate sequence stratigraphic interpretation methods using manual and automated means to compare the fit of results with pyBadlands Stratigraphic Forward Model (SFM) input and output.

Response: We added a new section (4.1) in which we quantified the timing and development of stratigraphic surfaces and depositional units based on the temporal evolution

of stratal stacking patterns (Page 9, Figure 5). The stratigraphic analysis in section 4.1 also serves as a reference for comparison with interpretations resulting from the two tested methods. We restated the three aims of our work in the Introduction, which are (1) use SFM as a tool to quantify the development of stratigraphic architecture under the interplay between accommodation change and sediment supply (Page 2, lines 19-20); (2) evaluate the performance of the trajectory analysis and the accommodation succession method on the interpretation of stratal architecture predicted with pyBadlands (Page 3, lines 6-7); (3) integrate quantitative stratigraphic analysis within pyBadlands based on the trajectory analysis and accommodation succession method (Page 3, lines 13-15).

Comment 1-2: The approach used is novel, applying different interpretation techniques on the output of a SFM and comparing the results of each technique against time-dependent SFM inputs and outputs. Unfortunately, there are flaws in the analysis that stem from a blurring of observations that are the foundation of interpretation methods and the forcing mechanism inferred to drive them. Firstly, a $\beta$-factor of 1 to 2.5 over 150 km produces a subsidence profile which increases so much toward the basin that 10 million year duration, 50m "eustasy" cycles don't produce basinward shifts of facies (depositional sequence boundaries) resulting from negative shelfal accommodation that is a key factor to interpretation with either shoreline trajectory (ST; Helland-Hansen and others '94-'09) or accommodation succession (AS; Neal and others '09-'16) methods.

Response: The imposed three cyclical eustatic sea level changes do induce progradation stacking, basinward shifts of facies and formation of subaerial unconformities during sea level fall. The average rate of sea level change is 10 m/Myr. The imposed thermal subsidence over the shelf (150 km to 250 km on a dip-oriented cross-section) has a subsiding rate ranging from 0 to 16.7 m/Myr. Therefore, the shelfal accommodation varies with the fluctuation of eustatic sea level. The imposed stretch factor $\beta$ is taken from McKenzie's model (1978) and is within a normal range for passive margin. We acknowledge that the prescribed increase in $\beta$-factor from 1 to 2.5 over 150 km

is large. We considered a test case with half the thermal subsidence. Similar progradation stacking, basinward shifts of facies and formation of subaerial unconformities are observed for this test case (see the figure below), in which the whole depositional package is accumulated $\sim 30$ km further basinward.

Comment 1-3: Application of ST method is disadvantaged as presented because the SFM produces a trajectory the authors had to invent ("descending transgressive trajectory class" or DTTC) in order to fit geometries with known sea-level conditions. This is a limitation to methods that are explicitly linked to sea-level change.

Response: Thank you for raising this issue. We understand the limitation of applying the shoreline trajectory analysis to our test case as the developed clinoforms are of shelf-lope scales over long-term (tens of millions of years) rather than shoreline scales (tens of thousands of years). We pointed this out in the Discussion (Page 16, lines 22-24). The numerical tools we provided to extract time-dependent shoreline positions based on a given sea-level forcing would also be useful for short-term SFM experiments.

Comment 1-4: The AS method explicitly avoids sea-level requirements and focuses on stratal terminations at key surfaces that bound different stacking patterns. This method allows interpretation to adjust to dipping strata that was initially horizontal (clinoform topsets – coastal plain aggradation). ST method builds from the assumption of trajectory from horizontal, so differentiating relative to AS is artificial (a function of forcing it to fit the sea level curve). THIS is the actual insight from Ding et al.'s paper – apply ST or AS methods but do not force them to fit a sea level curve. We don't observe sea level in stratigraphy, we infer it. We observe stratal terminations, shoreline trajectories, vertical and lateral stacking of facies associations, and key bounding surfaces that record significant changes in these observations.

Response: We agree with this comment and appreciate the objective characters of the AS method. When following the workflow of the ST and AS method to define different

stratigraphic units, we did not force them to fit a sea level curve. On the contrary, we clearly see the mismatches between the timing of stratigraphic surfaces and changes in sea level (Page 9, Figure 5), which are induced by basement subsidence and variations in sediment supply. We understand the importance of inferring sea level from stratigraphic record, and aimed to quantify the stratigraphic evolution for known forcing conditions (including sea level) to provide insights into reconstruction of contributing factors in natural systems. The fact that dA-dS is a proxy for the derivative of sea level change with respect to time, rather than a direct proxy for sea level change (see Comment 3 by Reviewer #1) is a reminder that sea level cannot be directly inferred from stratigraphic analysis.

Comment 1-5: The erosion feature of pyBadlands produces interesting 2D truncation geometries updip (but this was not demonstrated in the Wheeler diagram (fig. 5c) and might produce more interesting relations in shoreline trajectory if $\beta$-factor were reduced.

Response: Thank you for raising this interesting point. The truncation geometries are nearly horizontal when they are formed, and then evolves into upward dipping due to basement subsidence. The original Wheeler diagram was automatically constructed based on paleo-depth and therefore recorded instantaneous sediment deposition. Based on this comment, we added the final stratal thickness, which indicates the erosion of the progradational stacking, to the Wheeler diagram (Page 10, Figure 6b).

Comment 1-6: For scaling comparison, I suggest you refer to the physical flume model and resulting interpretation published in Martin et al. 2009 (Martin, J., Abreu, V., Neal, J. Sheets, B. 2009. Sequence stratigraphy of experimental strata under known conditions of differential subsidence and variable base level. AAPG Bulletin, 93, 503–533.)

Response: Thank you for referring us to this insightful paper, which we mentioned in the introduction (Page 2, line 13) and discussion (Page 15, line 26) of the revision.

The scalability of the Stratigraphic Forward Model would make it possible to carry out a scaling comparison to the work of Martin et al. (2009), however, such a comparison would distract from the message of this manuscript and it would require sufficient work to warrant a separate study

Comment 1-7: In summary, there are ways this experiment could be run that would make a better comparison of interpretation methods or the paper could more directly highlight shortcomings of interpretation methods that are explicitly linked to sea-level change. The approach in Ding et al. is innovative for using SFM to volumetrically quantify $\delta A/\delta S$ or ($\delta A$ - $\delta S$ if you wish) and I encourage the authors clarify their purpose (change the model or change the conclusions and application) so this good work is more on target.

Response: Thank you for this advice, based on which we clarified the aims of our work (see the reply to comment 1-1).
* * *
[Figure]

**(a)** 10 Myr

9.2    9.9
SU    4.0    6.5
CC*

**SU:** subaerial unconformity
**CC*:** correlative conformity (Hunt & Tucker, 1992)
**SB:** sequence boundary (SU+CC*) ————

| | |
|---|---|
| Alluvial plain | 0 |
| Shoreface | 10 |
| Distal offshore | 50 |
| Upper slope | 200 |
| Middle slope | 500 |
| Abyssal | |

Paleo-depth (m)

**(b)** 20 Myr

19.0    20.0
16.5
13.5

**CC:** correlative conformity ————
**MFS:** maximum flooding surface ————
**MRS:** maximum regressive surface ————
**#1 to #4:** depositional sequences

**(c)** 30 Myr

28.8    30.0
23.5    26.2
**4**
**3**
**2**
**1**

**1    #2    #3    #4**

0
-25
-50    → Sea level (m)

15
0
-15    → Rate of sea level
change (m/Myr)

0    5    10    15    20    25    30 (Myr)

Depth (m)

140    160    180    200    220    240
Distance (km)

**Fig. 1.** Figure 5

**(a)**

Depth (m): 0, 200, 400, 600

Stratal thickness (m): 0 10 20 30 40 50

**(b)**

Shoreface | Distal offshore | Upper slope

Time (Myr): 30, 25, 20, 15, 10, 5, 0

MFS
MRS
SB
CC
MFS
MRS
SB
CC
MFS
MRS
SB

Distance (km): 140 160 180 200 220 240

**Fig. 2.** Figure 6

[Figure]

**Fig. 3.** Supplementar figure to Comment 1_2. Predicted stratal stacking pattern from a new case forced with half of the original subsidence. Other forcing parameters remain the same.